# Oral Sources of Salivary Metabolites

**DOI:** 10.3390/metabo13040498

**Published:** 2023-03-29

**Authors:** Eelis Hyvärinen, Bina Kashyap, Arja M. Kullaa

**Affiliations:** Department of Oral Medicine, Institute of Dentistry, School of Medicine, Kuopio Campus, University of Eastern Finland, FI-70211 Kuopio, Finland

**Keywords:** saliva, gingival crevicular fluid, metabolites, oral disease, inflammation, oral microbiota

## Abstract

The oral cavity is very diverse, wherein saliva plays an important role in maintaining oral health. The metabolism of saliva has been used to investigate oral diseases as well as general diseases, mainly to detect diagnostic biomarkers. There are many sources of salivary metabolites in the mouth. Online English language sources and the PubMed database were searched to retrieve relevant studies on oral salivary metabolites. The physiological balance of the mouth is influenced by many factors that are reflected in the salivary metabolite profile. Similarly, the dysbiosis of microbes can alter the salivary metabolite profile, which may express oral inflammation or oral diseases. This narrative review highlights the factors to be considered when examining saliva and its use as a diagnostic biofluid for different diseases. Salivary metabolites, mainly small-molecule metabolites may enter the bloodstream and cause illness elsewhere in the body. The importance of salivary metabolites produced in the oral cavity as risk factors for general diseases and their possible relationship to the body’s function are also discussed.

## 1. Introduction

The metabolic analysis of human biofluids, including saliva, has progressed rapidly over the last decade evolving to obtain new biological information about different diseases and salivary biomarkers in diagnostics. Analysis of salivary metabolites has already become a common task in saliva research due to continuous development in analytical techniques, so that low-molecular-weight metabolites are easily accessible in different biofluids, including saliva. Mass spectrometry (MS) and nuclear magnetic resonance (NMR) spectroscopy are the most commonly used methods for the study of salivary metabolites [1]. Other less used methods in saliva research include Fourier-transform infrared (FTIR), photoacoustic spectroscopy, and Raman spectroscopy [2]. These new methods allow the same metabolites to be found in the saliva as in the blood, even though they are lesser quantitatively in the saliva. 

Whole-mouth (WMS) saliva is the most-used biofluid in the salivary metabolic analyses. The collection of saliva samples has many advantages because it is noninvasive, painless, and safe without piercing the skin. Saliva is easy to collect without complex equipment and does not require special expertise. Furthermore, the storage of saliva samples is simple and cheap. Saliva contains both endogenous and exogenous metabolites, which can tell us about biological pathways in the oral cavity. However, their role as biomarkers in the diagnostics of oral and systemic diseases is also investigated as reviewed by Hyvärinen et al. [3]. Saliva allows the measuring the levels of metabolites, including nucleic acids, lipids, amino acids, peptides, vitamins, organic acids, thiols, and carbohydrates, representing a useful tool for the detection of biomarkers for various oral diseases and monitoring disease progression. Salivary metabolites are involved in a variety of cellular functions, such as direct regulation of gene expression and they function as the effectors of molecular events that contribute to disease [4]. A minor part role for salivary metabolites could be as a potential source of systemic biomarkers, however, the main sources of salivary metabolites are productions of oral metabolic pathways, especially produced by microorganisms [5,6].

The oral cavity is a complicated organ with different niches for the colonization of millions of microorganisms including bacteria, fungi, viruses, protozoa, and archaea [7,8]. Saliva represents a major role in the maintenance of oral homeostasis with its multifactorial defense system and lubrication of different oral surfaces [9]. Oral inflammatory diseases and oral microorganisms contribute to the salivary metabolic fingerprint [5]. It is suggested that due to the high degree of connectivity in cellular metabolism, any disease will carry a metabolic signature that can be identified through the analysis of the metabolome in saliva and other tissue fluids. Furthermore, the multifunctional defense components of the saliva come from blood or serum via salivary glands, from gingival crevicular fluid (GCF) and epithelial cells.

Microorganisms play an important role in the metabolism of the oral cavity but it is not known how their diversity relates to salivary metabolites in patients with different oral or systemic diseases. However, the health of the oral cavity and oral diseases alter the saliva metabolism profile, and they must be considered in the investigations of systemic diseases. Hence, this narrative review aims to provide a general overview of salivary metabolites associated with the oral cavity and oral diseases. Metabolites produced in the oral cavity as risk factors for general diseases and their possible relationship to the body’s function are also discussed.

This narrative review is based on a literary search, which was performed on the PubMed database with the keywords “saliva”, “oral diseases”, “metabolites” and “oral cavity”. In addition, this article is based on the long-time clinical experience of two authors (BK, AMK).

## 2. Sources of Salivary Metabolites in Healthy Subjects

Saliva is an oral fluid secreted by the major and minor salivary glands. After entering the oral cavity, it is referred to as mixed or whole saliva, supplemented with many constituents originating from blood, mucosal cells, immune cells, and microorganisms [9]. Whole saliva represents a complex mixture of a variety of molecules and hence, it is valuable to use in research. The composition of saliva and oral microbiota differ in healthy subjects most likely due to age, gender, habits, diet, oral hygiene, medication, and different oral niches (prothesis, tooth fillings, tongue disorders, sebaceous glands = Fordyce granules) (Figure 1).

The oral microbiome is a set of diverse microorganisms that inhabit different niches of the oral cavity. They, together with salivary defense, play a key role in the oral balance between health and diseases. Oral microbes communicate with oral epithelial cells via Toll-like receptors (TLRs), which play a key role in the oral immune system producing inflammatory cytokines. Studies using scanning electron microscopy (SEM; Figure 1B) showed no or very few microorganisms on the buccal mucosal surface. Instead, most of the mucosal microorganisms are located on the dorsal tongue surface (Figure 1C), especially on the surface of rough hyphae of filiform papillae (Figure 1D).

GCF is another oral fluid and an inflammatory exudate derived from the blood vessels of gingival plexus, adjacent to the gingival epithelium. The bacterial degradation product in the GCF promotes the binding of calculus formation subgingivally [10]. GCF provides ease in collection and sampling of multiple sites of the oral cavity simultaneously due to its close interaction with periodontal cells and bacterial biofilm.

Fordyce granules (FG) are tubular–acinar sebaceous glands (Figure 1G), most often located in the lip and buccal mucosa and are more common in males. The ductus of FG opens into the oral cavity and a lipase-containing secretion is possibly passed into saliva. However, their significance in the salivary metabolic profile is likely to be limited. 

Dentures present different niches for the colonization of microorganisms (Figure 1H). Candidiasis without any symptoms is a quite common oral disorder in denture wearers. This also indicates dysbiosis of oral microbiota that further changes the salivary metabolic profile in denture-wearing patients. Furthermore, the denture can be colonized by respiratory pathogens, which can even be a risk of respiratory infection [11]. Individuals with appliances for orthodontic treatment are advised to practice proper oral hygiene. Failure in such practices results in plaque and calculus deposition superimposed with the bacterial degradation product causing gingival and periodontal inflammation. This further raises the possibility of changes in salivary metabolites. 

## 3. Salivary Metabolites in Patients with Oral Inflammation

The oral cavity contains a complex array of diverse microorganism that is tightly controlled by their host via metabolic machinery, substrate-specific or salivary secretory products. A mutually beneficial equilibrium exists between the host and oral microbiota until it is disturbed by external factors [12]. Several oral disorders, including caries, gingivitis, periodontitis, and oral ulcerations, relate to oral microbiota dysbiosis wherein generation of metabolites can result in inflammation-mediated tissue destruction (Table 1).

Caries and periodontitis are the most common oral inflammatory diseases in humans globally. Caries is more common in children and older people while periodontitis is the most common oral disease in the middle-aged population. Therefore, the age of individuals participating in a study may affect the metabolic profile. For example, in a comparative study of caries-related metabolites, a difference was found between adults’ and children’s salivary metabolites [21,22].

Both caries and periodontitis are induced by bacterial dysbiosis in the oral cavity. Dental caries results in an imbalance in the microbiota. Most of the microorganisms associated with tooth decay acquire a selective advantage over other species by changing the homeostatic balance of the salivary biofilm. The main source of carbohydrates for caries-causing microorganisms is consumed food. These carbohydrates usually leave the mouth within about an hour due to saliva’s lubrication effect of the saliva. Of course, this washout is affected by the saliva secretion rate, which means that hyposalivation in patients does not wash their mouth out in the same way as in subjects with normal salivary flow rate [23]. Microorganisms in the oral biofilm can metabolize dietary carbohydrates to produce organic acids, which will decrease pH and initiate the demineralization of dental hard tissue, developing caries [24]. High levels of free amino acids have been linked to increased protein hydrolysis activity by oral bacteria. High proline and glycine levels are the possible result of the hydrolysis of dentin-collagen in caries active individuals [25]. The high level of lipids on the salivary pellicle of tooth surfaces can inhibit the acid diffusion and accelerate to caries development [26]. Salivary metabolites produced by bacterial metabolism, including lactate, acetate and n-butyrate have been shown in patients with caries. These metabolites can reduce the pH and increase the porosity of the dental plaque matrix [27].

Periodontopathogenic bacteria contribute to periodontal diseases. The oral microbes release salivary metabolites as a product of multifactorial interactions between host, oral bacteria, and altered cellular metabolism of the host. The change in the metabolite concentration is correlated with the products of the pathogenic bacterial population. The change in the subgingival environment with regard to oxygen tension, redox potential, pH, and availability of host-derived macromolecules has been shown previously. Such changes are responsible in a cause-and-effect way for modulation of the bacterial composition [14] (Figure 2). Gingival crevicular fluid (GCF), whose composition is close to that of serum, flows into the oral cavity from periodontal pockets and varies in gingival inflammation [28]. Some anaerobic periodontal bacteria can produce very short chain fatty acid (SCFA) metabolites that are released from infection sites into the microenvironment. This can further contribute to the periodontal pathogenesis further through impairment of immune cells or fibroblasts and epithelial cell functions [29]. SCFAs, the end-products of bacterial metabolism such as butyrate, caproate, isocaproate, propionate, isovalerate and lactate have been linked to deep periodontal pockets, loss of insertion, bleeding, and inflammation. These metabolites are significantly decreased following periodontal treatment and gradually increase over time, which makes them possible indicators of periodontal disease development and progression [16].

In recent years, the relationship of periodontitis with systemic diseases such as cardiovascular diseases, diabetes mellitus, and problems during pregnancy, rheumatoid arthritis, chronic obstructive pulmonary disease, pneumonia, obesity, chronic kidney disease, metabolic syndrome and cancer has been amply demonstrated [30]. Several mechanisms have been proposed, including the transient bacteremia resulting in bacterial colonization in extraoral sites, systemic injury by release of free toxins and systemic inflammation triggered by soluble antigens of oral pathogens [31]. However, controversy exists between different studies due to the heterogeneity in the definitions and identification of periodontitis.

## 4. Salivary Metabolomics in Oral Mucosal Diseases and Oral Cancer

In the oral cavity, there are many kinds of tissue forming a very complex milieu. There are also many diseases in the mouth, of which only the most important are presented.

Oral ulcerations are the most common oral mucosal lesions. Recurrent aphthous stomatitis (RAS) is an inflammatory disorder, which is characterized by recurring and painful ulcers on the surface of the oral mucosa (Figure 2A). Typically, aphthous ulcers occur in adolescents and young adults, with the majority of patients affected being under 30 years of age and seldom in adults older than 40 years [32]. Ulcerations associated with aphthous stomatitis (RAU) are thought to represent a dysfunction of the oral immune system [33,34] Several studies have reported different etiology factors for RAU including the presence of certain oral microbial communities, immunological factors, endocrinopathies, and psychological and hereditary factors [35]. The difference in the oxidant/antioxidant status in the blood and saliva of patients with and without RAU has been previously described [36]). Only one study has been conducted on changes in the metabolite related to aphthous stomatitis [37]. An imbalance of tryptophan metabolism and steroid hormone biosynthesis has been shown to be correlated with increased incidence of oral ulcers [37]. Salivary metabolites have been shown to increase in serotonin, which influences psychological factors including depression and stress in patients with aphthous ulcers. However, the exact molecular mechanism remains unclear.

### 4.1. Tongue Disorders

Variation in color and structure of the tongue surface can be considered normal variation or caused by different diseases. These changes are easy to diagnose clinically because there are visible changes in filiform papillae (Figure 2G,H). Changes in these papillae may greatly alter the concentration of salivary metabolites, as oral microbes are harbored on these papillae normally in the healthy mouth. Geographic tongue (GT), also known as migratory glossitis, is a benign chronic inflammatory tongue condition. The typical clinical appearance is characterized by erythematous lesions with filiform papillae atrophy, surrounded by a white limited zone, producing a map-like appearance. Furthermore, fissured tongue (FT) is a disorder manifested as grooves and presents many enlarged, smooth papillae without hairs [38]. In patients with GT and FT, salivary calprotectin was elevated indicating the activation of neutrophils. Two opposite tongue lesions are hairy tongue (HT; Figure 2G) and atrophic tongue (AT; Figure 2H). On the surface of HT, there are a lot of long threads, onto which many microorganisms and food leftovers attach. Unlike HT, the filiform papillae have disappeared partially or completely in AT. It can be assumed that in both cases the biofilm of the tongue, like the whole oral microbial system, has changed.

### 4.2. Leukoplakia and Lichen Planus

Most of the oral cancers are preceded by asymptomatic or symptomatic clinical lesions with increased potential for malignant transformation, referred to as oral potentially malignant disorder (OPMD). Oral leukoplakia (OL) (Figure 2C) and oral lichen planus (OLP) are OPMDs [39]. Salivary metabolites studies have presented different metabolites in OL and OLP compared to oral cancer due to differences in pathophysiological stimuli (Table 2). The common elevated metabolites reported in OL are c- aminobutyric acid (GABA), phenylalanine, valine, lactate, eicosane, 4-nitroquinoline-1-oxide, and in OLP the metabolites are indole-3-acetate and ethanolamine phosphate. The metabolites of OL and OLP are shared with oral cancers suggesting their metabolic alterations are associated with the development of cancer. Increased glycolysis, products of glycolysis, carbohydrate metabolism, altered tricarboxylic cycle, phospholipid metabolism, lysine metabolic pathway and increased metabolic utilization are leading factors in the alteration in salivary metabolites[40,41,42,43]. This implies increased risk of OMPDs and oral cancers. As, salivary metabolomics provides a noninvasive approach for early detection of diseases by employing a novel and unique strategy, it can provide a complete metabolic response of organisms and their modification. However, such factors as the clinical appearance of OMPDs, age, gender, socioeconomic status, smoking and alcohol use might influence the salivary metabolites and must be considered in future studies.

### 4.3. Oral Cancer

The most common oral cancer is oral squamous cell carcinoma (OSCC). Despite current treatment methods, its five-year survival forecast is lower when compared to other cancers. Oral cancer is diagnosed worldwide every year and it causes the death in about 177,384 people per year [52]. Oral cancer is another condition to be considered when addressing the microbiome’s role in oral health. Most oral cancers are related to smoking habits and alcohol consumption; however, some studies reported other etiological factors, namely genetic susceptibility, external agents, and viral infections [53]. Recently, increasing evidence has indicated the role of bacteria in oral cancer. Oral dysbiosis was the term described in patients with oral cancer [54,55]. It was also suggested that metabolic bacterial products can be used as salivary markers for early detection of oral cancer [46]).

The mechanisms underlying the link between the oral microbiome metabolites and carcinogenesis are still unknown. It was shown previously that oral bacteria have the capacity to convert ethanol to acetaldehyde, which is a carcinogen [56]. Other proposed mechanisms suggest that the release of proinflammatory mediators that can disturb cellular cycling, disrupt signaling mechanisms, and act as tumor promoters [57].

The most frequently reported metabolites are amino acids (alanine, valine, leucine, threonine, proline, glutamic acid, phenylalanine, and choline), carbohydrates (N-acetylneuraminate (sialic acid), and the fatty acids (hexanoic acid and 2-hydroxypentanoate) (Table 2). The metabolites observed in oral cancer patients are related to the arginine and proline-, or cysteine and methionine-, or glycine and serine-, and glycerophospholipid and purine pathways [58]. Oral cancers are in direct contact with the saliva and tumor cell by-products are released into the saliva. Hence, the presence of hydrolyzed products by local tissue destruction or tumor cells in the saliva of oral cancer patients, enables identification of salivary metabolite identification.

## 5. Discussion

This article describes the most common oral factors that may affect the salivary metabolite profile. The challenge of salivary research is that both oral health and different habits and changes in human physiology affect the composition and volume of whole saliva.

It is well-known that many factors, including dietary behavior, oral hygiene, physical exercise, smoking, alcohol consumption and oral dysbiosis, influence oral health and metabolism in many ways. Salivary components attach and concentrate on the oral mucosal surface as a mucosal pellicle, which protects the oral mucosa from bacterial invasion and other irritants [59]. Hence, there are only a few microorganisms attached to oral healthy buccal mucosa (Figure 1B). Instead, the dorsal tongue surface contains many microorganisms on the surface of rough filiform papillae, which cover the entire tongue surface [38]. Morphological tongue changes, such as atrophic, hairy, geographic, and fissured tongue, can alter the oral biofilm considerably and further the salivary metabolic fingerprint. Investigation of the tongue surface may reveal much about the mouth and general health.

Oral diseases reduce well-being, and their role as a risk factor for many systemic diseases, including cardiovascular diseases, dementia, different cancers, and inflammatory bowel disease, have been shown in many studies. Further, periodontitis is involved in numerous systemic diseases, including cardiovascular disease, bacterial pneumonia, diabetes mellitus, dementia, and low birth weight [60,61,62]. The most extensive salivary studies on oral disorders have been conducted on the metabolism of periodontitis. However, the relationship between periodontitis and systemic diseases and their connection mechanisms is unknown.

The metabolic profile of the oral fluid was clearly identified for periodontitis. Some anaerobic bacteria from the supragingival and subgingival plaque can produce butyrate, one of the SCFAs. Butyrate is released from infection sites and contributes to the pathogenesis of periodontitis. In a chromatography analysis, the concentration of butyric acid was shown to decrease in patients with chronic periodontitis receiving periodontal treatment. However, gradually increasing concentrations of butyric acids were observed in the long-term after treatment, emphasizing the recolonization of periodontal pathogens [63]. The increase in the salivary butyrate, an important indicator of the pathogenic microorganism’s growth and periodontal tissue destruction was reported by some authors (Table 1). In four of the eight studies, salivary butyrate was elevated in patients with periodontitis, whereas one study showed a decrease in butyrate concentration after treatment [13,14,16,20]. Additionally, it was previously shown that with prolonged retention of butyrate within the gingival tissue, it will gradually enter the bloodstream [64]. Thus, butyrate related periodontal destruction could serve as a health risk factor capable of inducing systemic manifestations, which requires further elucidation.

Other SCFAs such as acetate, formate, propionate, caproate, isocaproate, propionate, isovalerate and lactate can also play important roles in periodontal pathology. These metabolites can promote an inflammatory response with the release of cytokines, hence, preventing repair at the cellular level [63]. Lactate, a product of lactic acid bacteria, forms a symbiotic relationship between bacteria and the host and can be metabolized to acetate and propionate [13]. In periodontitis, the tissue damage liberates different enzymes such as lactate dehydrogenase, related to cell death and destruction. Increased lactate levels in individuals with periodontitis have been reported [65]. Decrease in salivary lactate in periodontitis indicates a shift in the microbial composition of commensal bacteria in the oral cavity and other mucosal sites. Microbial species found between the teeth and the gums that inhibit the growth and colonization of exogenous pathogens have been recognized as beneficial [14]. Propionate levels can increase with an increase in the alpha-amino-3-hydroxy-5-methylisoxazole-4-propionic acid (AMPA) receptors that have a biological role in tumor invasion [66]. The rise in acetate levels in saliva indicate a restricted supply of carbohydrates and was correlated with caries in subjects [21]. Most oral cavity microorganisms are unable to utilize the carbohydrate energy sources, and this causes an increase in amylase activity and monosaccharide levels in periodontitis. The salivary amylase hydrolyzes starch to glucose, and high levels of glucose indicate excessive amylase activity in periodontitis [20]. 

Succinate, itaconate, fumarate and citrate, are the intermediate metabolites of the Krebs cycle. These metabolites can regulate immune responses and have a role in disease progression [67]. Succinate is synthesized within the mitochondrial matrix and is a well-described immunemetabolites. Succinate is an essential intermediate of the tricarboxylic acid (TCA) cycle that exerts pleiotropic roles beyond metabolism in both physiological and pathological conditions. The increase in succinate can induce secretion of inflammatory cytokines by activating the hypoxia-induced factor 1 (HIF-1) transcriptional pathway, which leads to inflammatory diseases’ development [68]. Succinate aggravates periodontitis through the succinate receptor, SUCNR1 [69]. In another study, a small-compound gel formulation was developed that specifically blocks SUCNR1 to prevent and treat periodontitis by inhibiting dysbiosis, inflammation, and bone loss [70].

These studies indicate that butyrate plays an important role in untreated periodontitis and may lead to a general low-grade infection throughout the whole body. The increase in butyrate and succinate and decrease in lactate are considered the salivary metabolic markers of periodontitis (Table 1). 

Generally, cancer cells utilize energy via the glycolysis pathway, and this results in high production of lactate in the body fluids. This is shown in many tumors as an increased level of lactic acid and such increased lactic acid is associated with a decrease in pyruvate [71]. Pyruvate is an essential component for entering the TCA cycle, but due to decreased pyruvate supply, the TCA cycle is supplemented with other amino acids such as valine, leucine, isoleucine, and phenylalanine. Leucine, isoleucine, and valine have important effects on protein synthesis, glucose homeostasis, and nutrient-sensitive signaling pathways. With an increase in the metabolic utilization in transformed cancer cells, increased or decreased variations in salivary metabolites were observed and reported in several studies (Table 2). In one study, the salivary metabolite gamma-aminobutyric acid (GABA), one of the TCA intermediates, was found at a decreased level, which correlates with an increased recurrence of OSCC [40]. Glutamine, another TCA intermediate, increases the utilization of GABA by cancer cells and decreases GABA levels in saliva; hence, it is regarded as an important diagnostic marker in some of the cancers [72,73].

Another study reported an increase in sphinganine-1-phosphate (SIP) and 4-nitroquinoline-1-oxide in OL and OSCC saliva. SIP is a known pro-inflammatory, promitogenic, and/or chemotaxic lipids that enables cancer progression [41]. It can interact with different receptors such as epidermal growth factor and transforming growth factor beta to accelerate carcinogenesis [74], while 4-nitroquinoline-1-oxide is a strong carcinogen known to elicit its carcinogenic potential by producing free radical oxidative damage to DNA [75]. Subsequent damage to DNA and oxidative stress affects the estrogen metabolism and estrogen related metabolites are either upregulated or downregulated in saliva [76]. The nucleotide biosynthesis pathway is also affected in OL and OSCC. Inositol 1,3,4-triphosphate (IP3R) is another important salivary biomarker in OSCC as IP3R plays an important role in growth, aggressiveness, and drug resistance via modulation of different signaling pathways [77].

Indole-3-acetate and ethanolamine phosphate were reported to be discriminating factors between OLP and OSCC [42]. The high level of salivary indole-3-acetate is correlated with its production by cancer cells, and it promotes tumor growth and progression. The intermediate of the phospholipid metabolism is ethanolamine phosphate which forms a main part of sphingolipids. Sphingolipids are phosphorylated and transferred as SIP, a known carcinogenic factor [41]. Hence, we can speculate that the heterogeneous nature of OPMDs and oral cancers produce various salivary metabolite profiles. Also, the complexity of systemic networks in the human body allows some communications between diseases and salivary glands that alter gene expression and protein metabolism, and this is reflected as salivary biomarker profiles.

A causal association has been established between alcohol consumption and oral cancers. One of the diagnostic markers for oral cancer is acetaldehyde (C_2_H_4_O). It is an oxidative form of ethyl alcohol and describes the metabolism of the oral cavity related to alcohol use. According to the International Cancer Research Institute (IARC), acetaldehyde is classified as a class I carcinogen, i.e., it has carcinogenic effects. Exposure to acetaldehyde affects oral microbial activity. Poor oral hygiene and smoking have also been shown to increase the concentration of acetaldehyde in saliva [78]. Understanding the metabolism of acetaldehyde in the oral cavity can give new information about oral metabolism related to oral cancer.

OSCC tumor cells are in contact with saliva. In this case, the metabolites produced by the cells via small vesicles (Figure 2E) are reflected in the saliva metabolism profile. Fucose (6-deoxy-L-galactose) is known to occur during the development of cancers, when the tumor cells modulate their surface by increasing fucosylation [79]. It has been shown that blood fucosis levels are high in different cancers [80,81], and also in oral cancers [82,83]. Only one preliminary study showed raised fucosis in saliva [84]. This relationship should be further investigated using a larger patient data set.

In the future, oral health will be one of the key factors to be considered in saliva metabolic studies. Therefore, the subjects to be examined should be selected according to oral health if the whole saliva is used for the determination of metabolites. There are also wide variations in different studies on salivary metabolites [85]. When collecting saliva for studies, certain factors should be considered such as timing (circadian rhythm), the stimulation of saliva flow, physical stress, and the abstention period of eating and drinking before the saliva collection. Other important factors include consistent methods of saliva collection and methods of analysis to assess the effects of general disorders on the saliva metabolism profile. A comparison between the different studies can be made after the same methods and a study protocol can be carried out. 

## 6. Conclusions

Oral health is one of the key factors to be considered in studies on salivary metabolomics. Oral bacteria and their metabolites, and inflammatory agents produced by the bacteria, may pass through microvessels from the invaded gingiva into the bloodstream, and then cause disease directly or indirectly. However, some of the salivary metabolites are transferred into the bloodstream via gingival pockets or via endocrine mechanisms of salivary glands communicating with other organs throughout the body.

Salivary metabolomics has a high potential for the diagnosis of systemic diseases. To validate the salivary metabolite data, a comparison between the different studies can be made when the same methods and a study protocol can be carried out. However, oral health condition should be taken into consideration in these studies.

## Figures and Tables

**Figure 1 metabolites-13-00498-f001:**
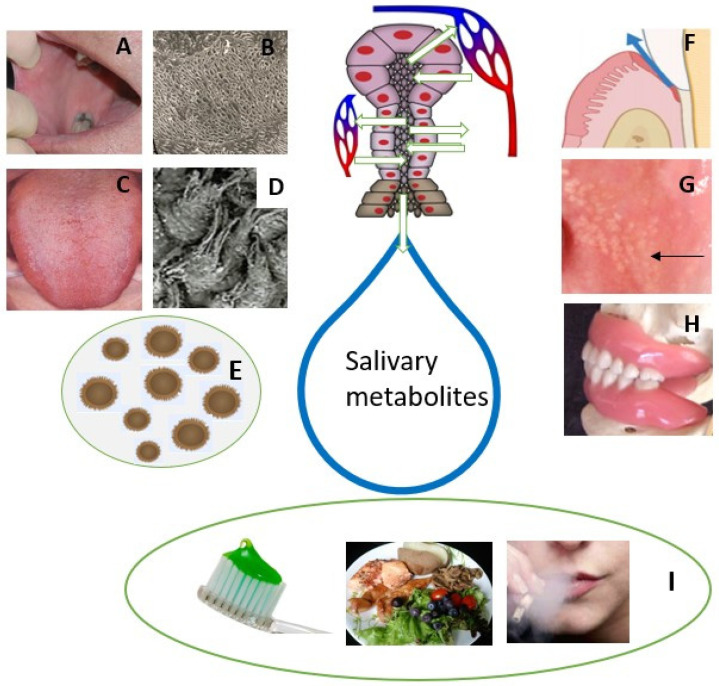
Summary of oral metabolite sources that may take part in salivary metabolic fingerprint in healthy subjects. Saliva is secreted by salivary glands in salivon with acinar and ductal cells. Metabolites are obtained also from oral mucosal cells (**A**–**D**) and gingival crevicular fluid (**F**). Most of salivary metabolites are produced by oral microorganisms (**E**). Sebaceous glands referred as *Fordyce granules* (arrow) are located in the buccal mucosa (**G**). Dentition and dentures (**H**) form different niches for attaching of microorganisms. Oral hygiene, diet, and habits, including smoking and alcohol consumption, differ salivary metabolite profile (**I**).

**Figure 2 metabolites-13-00498-f002:**
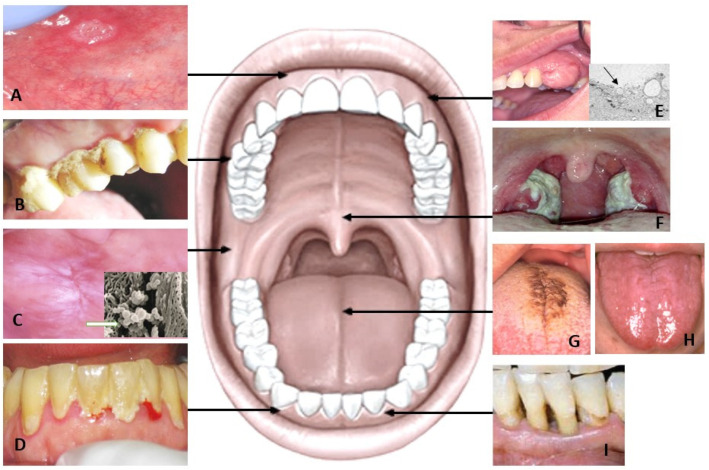
Aphthous ulcer presents erythematous halo surrounding round yellowish ulcer (**A**). Caries (**B**) is one of the most common individual illnesses. Leukoplakia of the buccal mucosa (**C**) can colonize more microorganisms than the healthy mucosa (studied with SEM). Dental plaque and bleeding of the gingiva (**D**) are sources of different metabolites in the whole mouth saliva. Oral tumors (**E**) and their small vesicles (arrow) produce metabolites, which can be used in diagnostics. Pharyngitis in tonsils (**F**) must also be remembered in saliva studies. Changes of the dorsal tongue surface, including hairy tongue (**G**) and atrophic tongue without hairs of filiform papillae (**H**) affect the biofilm of the oral cavity. Parodontitis (**I**) causes deep pockets around the teeth with many anaerobic bacteria altering the oral microbiome.

**Table 1 metabolites-13-00498-t001:** Salivary metabolites associated with most common oral inflammatory diseases, caries and periodontitis, based on NMR spectroscopic studies.

Oral Diseases N (HC/D) *	Saliva	Elevated Salivary Metabolites	Lowered Salivary Metabolites	Reference
Periodontitis 54 (22/32)	WS	acetate, c-aminobutyrate, n-butyrate, succinate, trimethylamine, propionate, valine	pyruvate, N-acetyl groups	[13]
Periodontitis 52 (26/26)	SWS	butyrate	fucose, lactate, acetate, N-acetyl of glycoprotein, GABA, 3-hydroxybutyrate, pyruvate, methanol, threonine, ethanol	[14]
Aggressive chronic periodontitis 100 (39/61)	USWS	proline, phenylalanine, isoleucine, valine, tyrosine	pyruvate, N-acetyl groups, lactate	[15]
Chronic periodontitis 130 (120/10)	USWS	caproate, isocaproate, butyrate, isovalerate, isoleucine, isopropanol, methanol, 4-aminobutyrate, choline, sucrose, glycine, lysine, lactate, proline		[16]
Chronic periodontitis, post-surgery 176 (52/124)	WS	lactate, ethanol, succinate, glutamate		[17]
Chronic periodontitis 45 (15/30)	US	propylene glycol, ethanol, lactate, acetoin, succinate, methanol, glycerol, formate	valine, propionate, isopropanol, alanine, pyruvate, acetate, acetone, choline, taurine, glycine, glucose	[18]
Periodontitis after therapy 23 (11/12)	USWS	alanine, glycine, taurine, proline, leucine, valine, isovalerate, tyrosine, methylamine, phenylalanine, isoleucine, lactate, formate, glucose, sarcosine, hypoxanthine, uracil	acetate, propionate, butyrate, ethanol, succinate, acetoin, galactose, aspartate, creatine, choline, methanol, pyruvate, isopropanol	[19]
Periodontitis 221 (92/129)	SWS	taurine, glucose, butyrate, isovalerate, glycolate, formate	ethanol, acetate, acetone, acetoin, choline, pyruvate, proline, lysine, propionate	[20]
Dental caries 30 (10/20)	USWS	butyrate, ambiguous, lysine, saccharide region, phenylalanine, and propionate.		[21]
Dental caries in children 38 (38/NM)	USWS SWS	alanine, aspartate, glutamine, glycine, isoleucine, leucine, proline, taurine, tyrosine, fucose, galactose, glucose, xylose, choline, dimethylsulfone, hypoxanthine, menthol, N-acetyls, uracil	butyrate, acetone	[22]

* N = total number of subjects; HC = healthy controls; D = diseased; NM = not mentioned; WS = whole saliva; USWS = unstimulated whole saliva; SWS = stimulated whole saliva; GABA = γ-aminoglutamate.

**Table 2 metabolites-13-00498-t002:** Salivary metabolites associated with oral mucosal lesions and oral squamous cell carcinoma (OSCC).

Oral Diseases N (HC/D) *	Saliva	Elevated Salivary Metabolites	Lowered Salivary Metabolites	Reference
OSCC 87/215	WS	alanine, taurine, pipecolic acid, leucine, isoleucine, histidine, valine, tryptophan, glutamic acid, threonine, carnitine		[44]
OSCC/OL 34/69	NM	alanine, lactic acid, 3-indolepropionic acid, n-eicosanoic acid	valine, proline, isoleucine, leucine, n-tetradecanoic acid, proline, threonine, phenylalanine, γ-aminobutyric acid	[40]
OSCC 30/30	USWS	lactic acid, hydroxyphenynactic acid, N-nonanoylglycine, 5-hydroxymethyluracil, succinic acid, ornithine, hexanoylcarnitine, propionylcholine	carnitine, 4-hydroxy-L-glutamic acid, acetylphenylalanine, spihingarine, phytosphingosine, S-carboxymethyl-L-cystein	[45]
Oral cancer 44/24	USWS	3PG, pipecolate, spermidine, met, SAM, 2 AB, trp, val, hypoxanthine, gly-gly, trimetrhylamine N-oxide, guanine, guanosine, tautine, choline, cadaverine, thr		[46]
OSCC 35/101	NM	glycine, proline, citrulline, ornithine		[47]
OSCC (OED, PSOML) NA/48	USWS	ornithine, carnitine, arginine, o-hydroxybenzoate, N-acetylglucosamine-1-phosphate, and ribose 5-phosphate (R5P)		[48]
OL and OSCC 18/43	USWS	d-glycerate-2-phosphate, estrone-3-glucuronide, 4-nitroquinoline-1-oxide, sphin-ganine-1 phosphate,1-methyl histidine, inositol 1,3,4-triphosphate, d-glycerate-2-phosphate, 2-oxoarginine, norcocaine nitroxide, pseudouridine	S-ureidoglycolic acid, p-chlorphenylalanine, d-urobilinogen, N-(3-Indolylacetyl)-l-isoleucine, tetradecanedioic acid, 1-hexadecyl hexadecanoate, l- homocysteic acid, ubiquinone, neuraminic acid, and estradiol valerate.	[41]
OSCC 124/249	USWS	putrescine, cadaverine, thymidine, adenosine, 5-aminopentoate	hippuric acid, phosphocholine, glucose, serine, adrenic acid.	[49]
OLP/OSCC NA/60	USWS	trimethylamine N-oxide, putrescine, creatinine, 5-aminovalerate, pipecolate, N-acetylputrescine, gamma-butyrobetaine, indole-3-acetate, N_1_-acetylspermine, 2′-deoxyinosine, ethanolamine phosphate, N-acetylglucosamine	N-acetylhistidine, o-acetylcarnitine	[42]
OLP 125/120	USWS	6 amino acid metabolites, 2 carnitines, 2 lipid metabolites and 9 other metabolites.		[50]
HNC with RT NA/9	USWS	histidine, tyrosine, urocanate, glycine, glutamic acid, aspartic acid, tryptophan, lysine, methionine, gamma-aminobutyric acid (GABA), butyrate, 2-isopropaylate, 2-aminobutyric acids		[51]
Oral cancer and OL 30/60	USWS	decanedioic acid, 2-methyloctacosane, Eicosane, Octane, 3,5-dimethyl, pentadecane, hentriacontane, 5,5-diethylpentadecane, nonadecane, oxalic acid, 6-phenylundecanea, L-proline, 2-furancarboxamide, 2-isopropyl-5-methyl-1-heptanol, pentanoic acid, docosane.		[43]

* N = total number of subjects; HC = healthy controls; D = diseased; NM = not mentioned; WS = whole saliva; USWS = unstimulated whole saliva; SWS = stimulated whole saliva; LC/MS = liquid chromatography-mass spectrometry; GC/MS = gas chromatography mass spectrometry; CE-TOF-MS = capillary electrophoresis time-of-flight mass spectrometry; UHPLC-MS/MS = ultrahigh performance liquid chromatography and tandem mass spectrometry; NMR = NMR-spectroscopy; OSCC = oral squamous cell carcinoma; OL = oral leukoplakia; OLP = oral lichen planus; PSOML = persistent suspicious oral mucosal lesions; HNC-head and neck carcinoma; RT-radiotherapy.

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
