# Peer review of "Oral Sources of Salivary Metabolites"

_metabolites, 2023, doi:10.3390/metabo13040498_

Round 1

Reviewer 1 Report

dear authors:

1- the abstract should contain the main methods and major findings, should avoid explaining the literature in the abstract

2-what methods did you use to find the papers you used in each section?

3- why the first part is "introduction" and after that you start numbering based on the findings you had

4-many headings have no headings number (like cancer,)

5-are the figures original? if no you must obtain the appropriate copyright

6-conclusion should be the main conclusion of your study

Author Response

1- the abstract should contain the main methods and major findings, should avoid explaining the literature in the abstract

Author response: We appreciate the reviewer’s comment. Authors have corrected the abstract and added the main method and findings.

2-what methods did you use to find the papers you used in each section?

Author response: Authors have made a narrative review by including the English language and Pubmed articles related to the oral salivary metabolites. Added to the text:

This narrative review is based on the literary search, which was made in PubMed database with the keywords "saliva", “oral diseases”, "metabolites" and "oral cavity". In addition, this article is based on the long-time clinical experience of two authors (BK, AMK).

3- why the first part is "introduction" and after that you start numbering based on the findings you had

Author response: We acknowledge the reviewers comment and we apologize for the improper numbering in the manuscript. Authors have corrected the mistake and the numbering has been made proper.

4-many headings have no headings number (like cancer,)

Author response: Authors have made the correction and provided number to each section.

5-are the figures original? if no you must obtain the appropriate copyright

Author response: Yes, the figures are original. Clinical images are taken by authors and are taken their previous clinical and research work as mentioned in the introduction.

6-conclusion should be the main conclusion of your study

Author response: As per the reviewers comment the conclusion has been modified.

Reviewer 2 Report

In the manuscript oral sources of salivary metabolites associated with the oral cavity and oral diseases has been reviewed. The subject matter of this review is important and interesting, especially regarding the aspect of the possibility of early diagnosis of diseases based on markers in saliva samples. One of the important advantages of analyzing saliva samples is their non-invasive collection. Of particular interest is the information collected in the Tables and the discussion about them.

However, the following corrections or suggestions may be addressed before publication.

The manuscript should be supplemented with a discussion of previous reviews on some similar subject matter e.g. Medicine in Novel Technology and Devices 13 (2022) 100115 https://doi.org/10.1016/j.medntd.2022.100115, J Clin Med. 2020 Feb; 9(2): 466. https://doi.org/10.3390/jcm9020466. The novelty and advantages of the manuscript compared to previously published articles should also be emphasized.

Information on how the literature was browsed should also be completed (e.g. what databases were reviewed, what time period was taken into account).

Some sections are incorrectly numbered. This should be corrected.

Author Response

In the manuscript oral sources of salivary metabolites associated with the oral cavity and oral diseases has been reviewed. The subject matter of this review is important and interesting, especially regarding the aspect of the possibility of early diagnosis of diseases based on markers in saliva samples. One of the important advantages of analyzing saliva samples is their non-invasive collection. Of particular interest is the information collected in the Tables and the discussion about them.

Author response: We are obliged with the reviewer’s comment.

However, the following corrections or suggestions may be addressed before publication.

The manuscript should be supplemented with a discussion of previous reviews on some similar subject matter e.g. Medicine in Novel Technology and Devices 13 (2022) 100115 https://doi.org/10.1016/j.medntd.2022.100115, J Clin Med. 2020 Feb; 9(2): 466. https://doi.org/10.3390/jcm9020466. The novelty and advantages of the manuscript compared to previously published articles should also be emphasized.

Author response: The article related to oral diseases has been added to the discussion.

 Information on how the literature was browsed should also be completed (e.g. what databases were reviewed, what time period was taken into account).

Author response: Authors have included the information regarding the narrative review in the abstract and Introduction sections.

Some sections are incorrectly numbered. This should be corrected. 

Author response: We appreciate reviewer’s comment. Authors have corrected the numbering in the manuscript.

Reviewer 3 Report

This is a well written review on the salivary metabolites and diseases.

On the whole, the ms is well written, comprehensive  and the contents are sound.

Some information on the proteomics and RT-PCR in saliva could be described.

Author Response

This is a well written review on the salivary metabolites and diseases.

Author response: We are grateful to the reviewer’s comment.

On the whole, the ms is well written, comprehensive and the contents are sound.

Author response: Authors are obliged with the comment.

Some information on the proteomics and RT-PCR in saliva could be described.

Author response: We appreciate reviewer’s comment. Authors have confined their review with the salivary metabolites studied with spectroscopies, mainly NMR. Hence, we could not involve other proteomics and RT-PCR studies.

Round 2

Reviewer 1 Report

Dear authors:

1-please creat a section named methods and explain your methodology in details

2-even the figures are from the published work of yourself, you need to take appropriate copyright 

Author Response

1-please creat a section named methods and explain your methodology in details

This narrative review is based on the literary search, which was made in PubMed database with the keywords "saliva", “oral diseases”, "metabolites" and "oral cavity". 

2-even the figures are from the published work of yourself, you need to take appropriate copyright 

 This article is based on the long-time clinical experience of two authors (BK, AMK).and the figures are not from published works.

Round 3

Reviewer 1 Report

Thank you very much